



# Slab Break-offs in the Alpine Subduction Zone

Emanuel D. Kästle[1], Claudio Rosenberg[2], Lapo Boschi[2,3], Nicolas Bellahsen[2], Thomas Meier[4], and
Amr El-Sharkawy[4,5]

[1]Institute of Geological Sciences, Freie Universität Berlin, Germany
[2]Institut des Sciences de la Terre, Paris (iSTeP), Sorbonne Université, Paris, France
[3]Dipartimento di Geoscienze, Universita' degli Studi di Padova, Italy
[4]Christian Albrechts Universität Kiel, Germany
[5]National Research Insititue of Astronomy and Geophysics (NRIAG), 11421, Helwan, Cairo, Egypt

**Correspondence:** E.D. Kästle (emanuel.kaestle@fu-berlin.de)

**Abstract.** After the onset of plate collision in the Alps, at 32–34 Ma, the deep structure of the orogen is inferred to have changed dramatically: European plate break-offs in various places of the Alpine arc, as well as a possible reversal of subduction polarity in the eastern Alps have been proposed. We review body-wave tomographic studies, compare them to our surface-wave-derived model for the uppermost 200 km, and reinterpret them in terms of slab geometries. We infer that the shallow subducting portion

of the European plate is likely detached under both the western and eastern (but not the central) Alps. The Alps-Dinarides transition may be explained by a combination of European and Adriatic subduction. This would imply that the deep, high-velocity anomaly (>200 km depth) mapped by tomographers under the eastern Alps is a detached segment of the European plate. The shallower fast anomaly (100–200 km depth) can be ascribed to European or Adriatic subduction, or both. These findings are compared to previously proposed models for the eastern Alps in terms of slab geometry, but also integrated in a

new, alternative geodynamic scenario that best fits both tomographic images and geological constraints.

## 1   Introduction

The Alps and the adjacent Apenninic, Dinaric and Carpathian mountain chains were formed by a complex history of subductions, collisions and backarc extension. During N-S convergence of Africa and Europe (e.g., Dewey et al., 1989; Stampfli and

Borel, 2002), between 84 Ma and 35 Ma, Adria-Africa acted as upper plate along most of the subduction front and the Alpine Tethys (Europe) was subducted towards the south (e.g., Schmid et al., 2004). An opposite subduction direction only affected what would later become the Dinaric-Hellenic arc (Laubscher, 1971; Pamić et al., 2000; Schmid et al., 2008; Ustaszewski et al., 2008, Fig. 1) and possibly also the southern Apennines (Argnani, 2012; Malusà et al., 2015). During the transition between subduction and collision (50–35 Ma, e.g., Handy et al., 2010; Carminati et al., 2012), slab break-offs are inferred to have

occurred in the Alpine and Dinaric collision zones (e.g., Blanckenburg and Davies, 1995; Lippitsch et al., 2003; Harangi et al., 2006; Kissling et al., 2006), partly preceding reversals of subduction polarity, as in the northern Apenninic collision (Vignaroli





et al., 2008; Handy et al., 2010; Molli and Malavieille, 2011) and possibly in the eastern Alps (Schmid et al., 2004; Handy et al., 2015). In this area, tomography results (Lippitsch et al., 2003) triggered a discussion on the apparent northward dip of the slab between 50 and 250 km depth, suggesting that it may result from subduction of Adria under Europe (Schmid et al., 2004; Ustaszewski et al., 2008; Schmid et al., 2013; Handy et al., 2015), hence to a change in subduction polarity, after the

onset of collision. This change would have been possible after the proposed break-off of the European slab, giving way to a northward indentation of the Adriatic crust and a subduction of its mantle (e.g., Schmid et al., 2013; Handy et al., 2015).

At present, the high-resolution regional tomographic models of the Alpine upper mantle disagree on important structures, such as the suggested present-day detachment of the European slab under the western Alps and the subduction polarity under the eastern Alps (Adriatic vs. European subduction) (Piromallo and Morelli, 2003; Lippitsch et al., 2003; Koulakov et al., 2009;

Dando et al., 2011; Mitterbauer et al., 2011; Zhao et al., 2016; Hua et al., 2017). An important limiting factor in the assesment of the structure of the Alpine slabs is the poor resolution in the uppermost mantle layer of teleseismic body-wave models (source-station distance $>30°$, due to almost vertical ray paths and lower data coverage (e.g., Boschi et al., 2010). In the light of a new model which does not suffer from this limitation (Kästle et al., 2018), we discuss the Alpine slab geometries and their possible detachments. The latter model is reliable down to a depth of approximately 200 km with decreasing resolution towards

depth due to the volume averaging properties of surface waves and the reduction in data coverage. Consequently, we interpret both the model from our surface-wave study and different body-wave models, combining the strengths of both approaches. The results are illustrated by four orogen-perpendicular cross sections that allow one to systematically compare the different tomographic models. Such a detailed side-by-side comparison cannot currently be found in the literature; yet, it is a necessary step to establish what we really know about the mantle structure under the Alpine belt. We interpret and discuss surface-wave

and body-wave models, while also taking into account the constraints provided by estimates of crustal shortening.

## 2 Tomographic imaging

While including information from different disciplines, the focus of this work lies in comparing and interpreting tomographic images derived from different studies. However, the comparability of models obtained with different methods and different datasets is not straightforward. We show several models based on body waves (Piromallo and Morelli, 2003; Lippitsch et al.,

2003; Koulakov et al., 2009; Dando et al., 2011; Mitterbauer et al., 2011; Zhao et al., 2016; Hua et al., 2017) and a recent one based on surface waves (Kästle et al., 2018). The imbalance between surface- and body-wave models in this comparison is explained by the fact that it has only been possible after the recent densification of seismic arrays to use surface waves to image the relatively 'small-scale' slab geometries underneath the Alps. The difficulty in comparing models lies, however, not only in using different wave-types but also in differences in datasets (number and geometry of station networks), data corrections

(e.g. correcting for crustal structure or elevation), data preparation (e.g. filtering, picking of P-/S-phases), data quality (strict or loose rejection of noisy data) as well as technical choices of the researchers such as model regularization (damping and smoothing), parameterization (grid spacing/cell size) or the underlying physical model (e.g. wave propagation as straight rays, bent rays or sensitivity kernels). All of these will finally influence the resolution of a model and consequently it is not possible



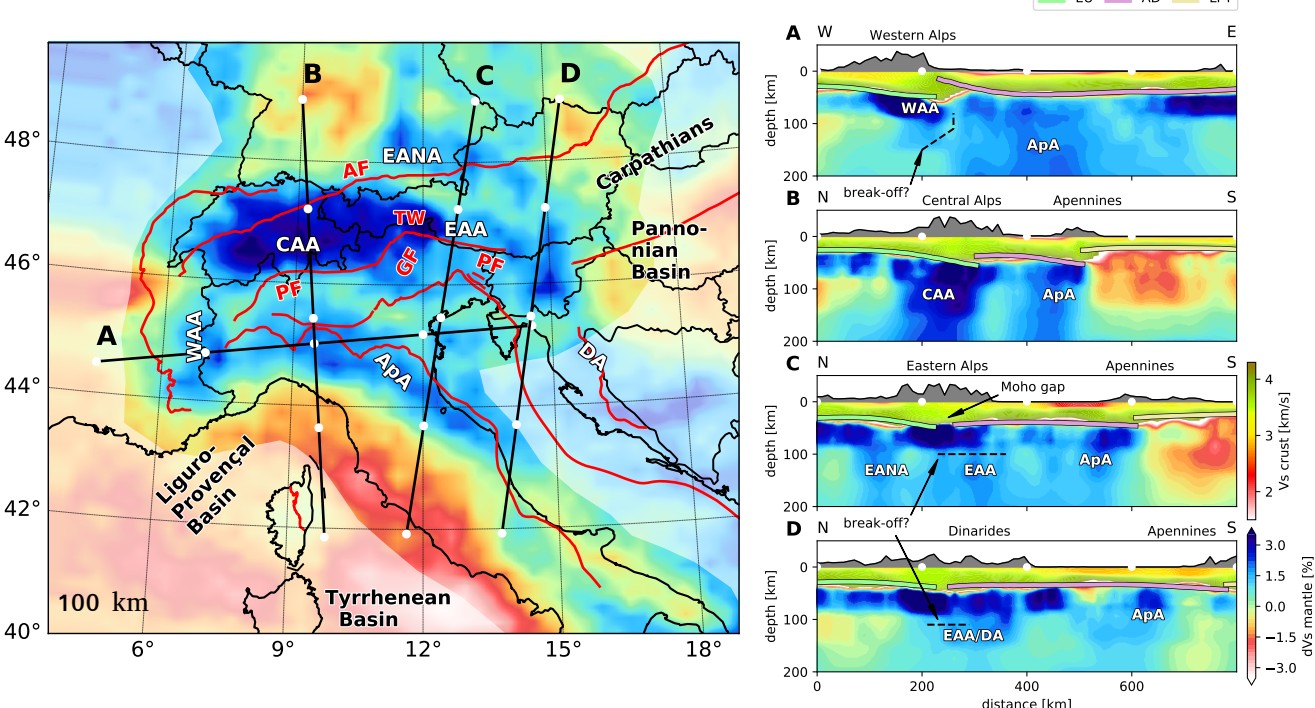

**Figure 1. Map:** Shear-velocity model from Kästle et al. (2018), showing deviations with respect to PREM (Dziewonski and Anderson, 1981). AF: Alpine frontal thrust, PF: Periadriatic fault, GF: Giudicarie fault, TW: Tauern Window. Faults and tectonic limits (red lines) simplified from Schmid et al. (2004, 2008); Handy et al. (2010). The white area marks the low-resolution region of the model. **Cross-sections:** Absolute shear velocity in the crust and PREM-deviations in the mantle. The dashed lines show the discussed levels for a potential European slab break-off. Section A is parallel to the anomaly under the northern Apennines. Moho boundaries from Spada et al. (2013). EU: Europe, AD: Adria, LPT: Liguro-Provençal and Tyrrhenian basins. Mantle anomalies: ApA: Apenninic Anomaly, CAA: Central Alpine Anomaly, DA: Dinaric Anomaly, EAA: Eastern Alpine Anomaly, EANA: Eastern Alpine Northern Anomaly, WAA: Western Alpine Anomaly.

to absolutely quantify the uncertainties for each of the models. For example, the imaged amplitudes of the velocity variations in the subsurface vary significantly between different models although the underlying dataset may be quite similar (e.g., Piromallo and Morelli, 2003; Koulakov et al., 2009) – for this reason we decide to use variable color scale limits for the analyzed models in the present study. These uncertainties make it even more necessary to use a series of models instead of a single one when trying to interpret the imaged structures in a tectonic context. In order to provide the reader with the first order differences between the models, we summarize their key parameters in Table 1; for a more detailed discussion of the models and an estimate of their resolution, the reader is referred to the original publications.





**Table 1.** Comparison of the key parameters for the models discussed in the text. Note that the parameter values are not a measure for the resolution or quality of the models. Resolution depends additionally on the data quality, geometry of the data coverage and quality of the corrections. Teleseismic corresponds to source-station distances $> 30°$.

| Author | no. stations | no. events | no. measurements | hor. grid spacing | crustal corrections | coverage |
|---|---|---|---|---|---|---|
| Piromallo and Morelli (2003) | >300 (Alpine area) | ~52,000 | 52,514 regional, 59,625 teleseismic (after ray averaging) | 0.5° | no correction, but regional events | entire Europe |
| Lippitsch et al. (2003) | ~200 | 76 | 4,199 | 50 km | regional crustal model, including 3D propagation effects | Alps |
| Koulakov et al. (2009) | ~200 (Alpine area) | >1,000 | 40,000–720,000 (area dependent, regional and teleseismic) | variable (>30 km) | European crustal model | entire Europe |
| Dando et al. (2011) | 100 | 225 | 23,869 | 25 km | different local models | parts of the Alps |
| Mitterbauer et al. (2011) | 154 | 80 | 6,368 | 30 km | regional crustal model, including 3D propagation effects | parts of the Alps |
| Zhao et al. (2016) | 527 | 199 | 41,838 | 25 km | regional crustal model | Alps |
| Hua et al. (2017) | 667 | 12644 teleseismic, 4014 regional | 439,386 teleseismic, 117,885 regional | 0.5° | crust inverted for with local events, Moho depth from European crustal model | Alps |
| Kästle et al. (2018) | 511 | - | ~90,000 | 10 km | crust inverted for with ambient noise | Alps |

## 3 Surface-wave Tomography

Surface-wave phase velocities are particularly sensitive to the vertical changes in shear-wave velocities, therefore they are well suited for probing the lithosphere and the asthenosphere. In a recent study, Kästle et al. (2018) combined phase-velocity-dispersion measurements from ambient-noise and earthquake data which resulted in a very broad frequency range (4 -300 seconds) of measurements, and an unprecedentedly dense ray-path coverage of the region. From the inversion of the phase-velocity measurements, we constructed a high resolution 3D shear-wave-velocity model underneath the Alpine region, showing that a 100 km thick anomaly (as from a lithospheric slab) can be identified down to 200 km depth (Fig. 1). The shear-velocity model of Kästle et al. (2018), shown in Fig. 1, thus provides an unique opportunity to study the crust-mantle transition and the approximate position and geometry of subduction slabs, under the Alps and adjacent orogens, despite the absence of deep earthquakes in this region.

The positions of the anomalies are subject to increasing uncertainty with depth, meaning that the surface-wave model is not well suited to estimate the slab dip. This may especially cause differences in anomaly positions with respect to body wave models. Also smearing and weakening of the imaged anomalies is observed at depth greater than 100 km caused by the large wavelengths of surface waves and the reduction in data coverage at long periods in the dataset (Kästle et al., 2018). Given that we use a pre-AlpArray (Hetényi et al., 2018a) dataset, there is also considerable variability in station coverage which can additionally cause lateral variations in anomaly strength. Areas with sparse station coverage tend to show reduced anomaly strengths and vice versa. This may enhance the anomaly for example in Switzerland (Fig. 1), where the station coverage is



denser. To ensure that these uncertainties do not affect our interpretation, we evaluated and tested two independent datasets (from ambient noise and from earthquake data) before merging them into the model shown in Figure 1 (Kästle et al., 2016, 2018). Resolution tests that show how the station coverage affects the recovered anomalies are also shown in Kästle et al. (2018).

The clearest velocity anomaly is obtained from the thick, vertical European slab under the central Alps (Fig. 1). In contrast, the Adriatic slab underneath Apennines and Dinarides is associated with a weaker high-velocity anomaly, which may be due to a thinner slab, or to a lower resolution, because the station distribution is not homogeneous, i.e. very linear within Italy and very sparse along the Dinarides.

**Central Alps:** The strongest and most continuous high-velocity anomaly is found under the central Alps. It extends from the
bottom of the crust down to the lower boundary of the mapped region at 200 km depth, suggesting a continuous slab of at least 100 km length (Fig. 1B). The steeply dipping slab anomaly extends in E–W direction from 7° to around 12.5°, with a thickness of up to 100 km.

**Western Alps:** In contrast to the central Alps, there is no clear, continuous high-velocity anomaly under the western Alps in our shear-velocity model (Fig. 1A). The fast, lithospheric velocity anomaly under the western Alps (European) vanishes at
depths below 90 – 120 km. A broader, fast anomaly is visible under the Apennines as section 1A cuts parallel to the orogen and the supposed position of the Apenninic slab (compare sections A and B in Fig. 1). The shallow western Alpine (European) and the Apenninic (Adriatic) fast anomalies are separated by a low-velocity anomaly, suggesting the presence of a shallow European slab break-off.

**Eastern Alps:** The central Alpine anomaly extends eastward to 12–13°E, coinciding approximately with the Giudicarie fault
(Fig. 1). At this longitude, we observe a sharp change in the amplitude of the high-velocity anomaly: further east, the strong, fast anomaly disappears below about 80 km in our surface-wave model (Fig. 1C,D). Below this depth, a region of more diffuse, slightly elevated velocities (Fig. 1C) and no clear separation between eastern Alpine and Dinaric domain is observed. In addition, a separate high-velocity anomaly (EANA, profile 1C) is found more to the north beneath the European foreland at depth of about 100–150 km. The reduction of anomaly strength suggests that the European slab under the eastern Alps
must be thinned, or even broken off. Alternatively, the high-velocity anomaly under the eastern Alps at 60 – 200 km depth may correspond to an Adriatic slab, which generally shows a lower velocity anomaly strength underneath the Apennines in our model. The imaged anomaly strength may furthermore be influenced by the reduced station density in the eastern Alps compared to the central Alps, however, resolution tests show that this effect should be small (Kästle et al., 2018). There is a small increase in anomaly strength towards the Dinarides. Under the northern Dinarides, we interpret the anomaly attaining
approx. 150 km depth to be of Adriatic origin (map view and *DA* in section 1D).

## 4  Body-wave Tomography

Using the tomographic models kindly provided by their authors (Piromallo and Morelli, 2003; Lippitsch et al., 2003; Koulakov et al., 2009; Dando et al., 2011; Mitterbauer et al., 2011; Zhao et al., 2016; Hua et al., 2017), we define 4 section traces and



**Figure 2.** Comparison of body-wave tomographic models along four cross-sections. The models show compressional velocity anomalies with respect to PREM (Dziewonski and Anderson, 1981, *sp6* in case of Piromallo and Morelli, 2003). The absolute anomaly strengths may vary between models due to the chosen color scale and model uncertainties (see text). The abbreviations in the cross-sections indicate our reinterpretations of the provenance of the slab anomalies, either European (EU), Adriatic (AD) or unclear/both (EU/AD). AF: Alpine frontal thrust, PF: Periadriatic fault, GF: Giudicarie fault. Faults and tectonic limits (red lines) simplified from Schmid et al. (2004, 2008); Handy et al. (2010). The crustal segments are attributed to the different domains according to Spada et al. (2013). EU: European plate; AD: Adriatic plate; LPT: Liguro-Provençal and Tyrrhenian domains. The white band in the Lippitsch et al. (2003) sections marks the shallow region which is not shown in the original model.





plot these sections from each model, applying the same velocity color model to each model (Fig. 2), hence allowing for an easy visual comparison. The labels on the fast anomalies in Figure 2 show our reinterpretations of the slab provenances as discussed in the text and are not necessarily in agreement with the original interpretations of the model authors. For the sake of clarity, we selected only three models for Figure 2 and present all other models in the supplementary material (Figs. S2–S12). The model

of Lippitsch et al. (2003) was selected, because it was the first one to image the Alpine slabs in cross sections and to suggest a slab break-off under the western Alps and a polarity reversal under the eastern Alps; the model of Koulakov et al. (2009), because it covers much of Europe, allowing interpretations that go beyond the Alpine region; the model of Zhao et al. (2016) is one of the most recent ones and was the first to propose continuous slabs all along the Alpine arc. These models also illustrate the differences between different data sets and methods (Table 1). The amplitudes of a given anomaly can differ significantly

due to data coverage and inversion parameters. We therefore concentrate primarily on the geometries of the anomalies rather than amplitudes.

**Western Alps:** In the western Alps, Lippitsch et al. (2003) propose a slab break-off at lithospheric depth (Fig. 2A). Such a break-off is also proposed by the model of Beller et al. (2017), using full-waveform inversion along a dense seismic profile. However, others observe a continuity between the deeper anomaly and the lithospheric one (Koulakov et al., 2009; Zhao et al.,

2016; Hua et al., 2017). At depths greater than 150 km, all models agree on a broad, fast anomaly (Fig. 2), which may represent a combination of European and Adriatic (from the Apenninic subduction) slabs.

**Central Alps:** All body-wave models show a continuous slab to at least 200 km depth under the central Alps (Piromallo and Morelli, 2003; Lippitsch et al., 2003; Koulakov et al., 2009; Mitterbauer et al., 2011; Zhao et al., 2016; Hua et al., 2017). Below this depth, the slab is thinned (Lippitsch et al., 2003; Mitterbauer et al., 2011; Hua et al., 2017), possibly indicating a break-off.

It may also be merged with the Apenninic slab (Fig. 2B), which would be indistinguishable from a continuous slab to ≥400 km depth. Such a long continuous slab has been proposed by Zhao et al. (2016). Towards the east, the slab anomaly is limited in some models by a slab gap (Lippitsch et al., 2003; Koulakov et al., 2009) or a northward step of the anomaly (maps in Fig. 2, Fig. S1–S3, Zhao et al., 2016).

**Eastern Alps:** An eastward increase in the northward slab dip by Lippitsch et al. (2003), while the other models (Piromallo

and Morelli, 2003; Koulakov et al., 2009; Dando et al., 2011; Mitterbauer et al., 2011; Zhao et al., 2016; Hua et al., 2017) show a sub-vertical, though slightly northdipping slab (Fig. 2C,D, S10–S12). Lippitsch et al. (2003) image the slab down to a depth of about 250 km (Fig. 2D). Dando et al. (2011) indicate that the slab may be more continuous (Fig. S10–S12) and linked vertically to a deep anomaly (>410 km) under the Pannonian Basin. A continuation of the fast anomaly down to at least 350 km, is also visible in models of Koulakov et al. (2009), Mitterbauer et al. (2011), Zhao et al. (2016) and Hua et al. (2017)

(Figs. 2C,D, S10–S12). The latter shows almost no fast anomaly in the upper 200 km, but a clear anomaly below, opposite to the model of Lippitsch et al. (2003) (Fig. 2D). Horizontal sections at 150 km (Fig. 2, S1) show that the eastern Alpine anomaly is separated from the central Alpine one by a slab gap (Lippitsch et al., 2003) or a northward step, spatially coinciding with the Giudicarie fault (Mitterbauer et al., 2011; Zhao et al., 2016).

An additional fast anomaly is present north of the Alpine Front in the model of Lippitsch et al. (2003), whose maximum

strength is at approximately 150 km depth, similar to the EANA in the surface-wave model (Fig. 1).





At the transition between eastern Alps and Dinarides (Fig. 2D), Lippitsch et al. (2003) indicate a shallow anomaly (∼100 km) under the northern Dinarides and a deeper one (∼200 km) under the eastern Alps. These two anomalies seem to be laterally linked, indicating that the Adriatic slab from the northern Dinarides might extend into the area underneath the eastern Alps. A similar link between northern Dinaric and eastern Alpine slabs is imaged by Dando et al. (2011) (Fig. S12). In the other models

the connection is much weaker or not present, however, they all show a deep anomaly, located below 300 km and potentially detached from the lithospheric anomalies (Figs. 2D, S12 at 200 km distance; Piromallo and Morelli, 2003; Koulakov et al., 2009; Mitterbauer et al., 2011; Zhao et al., 2016; Hua et al., 2017). The model of Hua et al. (2017) is partially opposed to the other models showing a low velocity anomaly at shallow depth, which may indicate that there is a zone without slab between 100–200 km depth possibly showing a break-off. In the other model such a clear low velocity zone appears only towards the

eastern limit of the Alps at the border to the Pannonian basin (Figs 2D, S12). In some models this limit is found further east of our easternmost section D (Fig. S3, Dando et al., 2011).

## 5 Discussion

### 5.1 Western Alps

Based on our surface-wave model (Fig. 1A), the European slab under the western Alps shows a shallow break-off at approxi-

mately 100 km depth. This agrees with the models of Beller et al. (2017) (full-waveform modeling) and Lippitsch et al. (2003, body-wave tomography), who locate the gap between 80 and 150 km depth. In contrast, other tomographic models image a continuous slab down to at least 250 km depth (Koulakov et al., 2009; Zhao et al., 2016; Hua et al., 2017; Lyu et al., 2017), which may imply that there was no break off at all in the western Alps.

Although the resolution is limited, it is clear that the shallow attached European lithosphere and the subducted Adriatic litho-

sphere beneath the northern Apennines are located very close to each other and may already be in direct contact (Zhao et al., 2016). Lippitsch et al. (2003) interpret the fast anomaly as the detached European slab, dipping towards the east. The model of Zhao et al. (2016) provides indications that the western European Alpine slab and the Adriatic slab from the northern Apenninic subduction may be merged below 200 km (Fig. 1. This may influence the apparent northward dip of the combined slab under the northernmost part of the Apennines. Also in other models, there is no visible separation between European and Adriatic

slab (Koulakov et al., 2009; Hua et al., 2017, Fig 2). Our surface-wave model, below approx. 100 km, can only resolve a single anomaly under the Apennines that we attribute to the Adriatic slab, because of its location and continuity with the lithosphere under the Apennines.

The collisional shortening estimates for the western Alps range between 30 – 150 km, when taking the shortening of the European units and 50 km of potential underthrusting of Europe into account (Bellahsen et al., 2014; Schmid et al., 2017; Rosenberg

et al., 2019). The inferred amount of shortening is lowest in the south, increases to about 100 km along our cross-section A (Fig. 2) and reaches its maximum in the north at the transition to the Central Alps.

Assuming the break-off scenario, its timing is an important, but still open question. The shallow depth level of the slab gap imaged in several models (Lippitsch et al., 2003; Kästle et al., 2018; Lyu et al., 2017) could indicate a rather young age (few





Ma). This would fit to observations of current uplift (Escher and Beaumont, 1997; Nocquet et al., 2016) and exhumation in the western Alps that is likely not only related to deglaciation and erosion, but is inferred to have a mantle source (Fox et al., 2015; Sternai et al., 2019). This implies that a remnant of the broken-off part of the slab should be hanging in the upper mantle, as indicated by the geometry of the fast anomaly in some body-wave models between 150-300 km depth (Fig. 2A, S4–S6).

This hypothesis does not exclude that the rupture took place at the ocean-continent transition, assuming that there may be a considerable delay of up to 20 Ma between collision and break-off (van Hunen and Allen, 2011). Alternatively, the slab may have weakened by thermal erosion caused by hot surrounding asthenosphere. Models show that the interaction between slabs, as inferred for the European and Adriatic ones, can facilitate break-off (Király et al., 2016).

However, the break-off may have occurred much earlier, without a time delay, shortly after the onset of continental collision

around 35 Ma. This would be in agreement with the very large exhumation rates between approx. 35 and 25 Ma (Malusà et al., 2011, and references therein). Other indirect evidence for a slab break off such as an increase in sedimentation rates (Garzanti and Malusà, 2008) or a significant reduction in converge rate (Handy et al., 2010) suggest a time for the break-off event between 30–25 Ma. This could still give a shallow depth level for the break-off in current images, assuming a very low amount of European subduction in the western Alps since then. The inferred amount of shortening of 100 km along our section A since

35 Ma fit better with the scenario of a recent break-off, but would be in the range of uncertainty of the imaged slab lengths (Fig. 1A, 2A, S4–6). The absence of a slab is shown clearest in the southernmost sections (Fig. S4) but is more ambiguous towards the north, which could be related to the increasing amount of shortening from south to north. In the scenario of a rather early (35 Ma) break-off, we assume that the broken-off part of the slab should be at least at 300 km depth, taking low sinking velocities of 1 cm/a into account (Capitanio et al., 2007; Replumaz et al., 2010). Contrarily to the previous scenario, this im-

plies that the fast anomaly at about 150-300 km depth in cross-section A (Fig. 4, labeled EU/AD) should be of purely Adriatic origin. This contradicts several model interpretations (Lippitsch et al., 2003; Zhao et al., 2016; Hua et al., 2017), however, the images along the western Alpine cross sections (Fig. 4A, S4–6) are not conclusive enough to finally discard the possibility that the anomaly is only showing Adria.

From a tomographic point of view, this shows that next to imaging the presence or absence of a slab break-off, it is also impor-

tant to better understand the geometry of the deeper lying slab(s) in order to be able to better distinguish between the different scenarios.

## 5.2 Central Alps

The central Alpine slab appears as a thick and almost vertical anomaly in all models down to at least 200 km depth (Fig. 1B, 2B). Only one of the shown models shows a continuous slab along all sections, suggesting an unbroken European slab from

the European subduction (Zhao et al., 2016). The vertical termination of the slab around 200–250 km depth in the other models (Piromallo and Morelli, 2003; Lippitsch et al., 2003; Koulakov et al., 2009; Mitterbauer et al., 2011; Hua et al., 2017), points to a slab length of approximately 100–150 km, closely matching interpreted amounts of convergence since 35 Ma (Schmid et al., 2004). The portion of inferred European shortening (63 km: Schmid et al. (1996), 30–95 km Rosenberg et al. (2015)), suggests that break-off must have happened at a depth greater than 100 km to explain the slab depth of ≥200 km. Recent results propose,


however, that the shortening in the European plate may be in the range of 150 km (Rosenberg et al., 2019). We conclude that the fast anomaly in the top 200 km is caused by syn-collisional subduction of the European plate. Below this depth, a break-off is likely, given the lack of a clear signal in the body-wave images. The deeper anomalies shown by Koulakov et al. (2009), Zhao et al. (2016) and Hua et al. (2017) are roughly located under the Po-basin and could represent the broken-off European

slab, or alternatively the Adriatic slab from the Apenninic subduction.

## 5.3 Eastern Alps

The classical interpretation (e.g. Hawkesworth et al., 1975; Lüschen et al., 2004) of a south-directed oceanic subduction followed by a south-directed continental subduction of Europe below Adria was questioned by Lippitsch et al. (2003), who

first argued in favor of a continental subduction of Adria below Europe, based on the imaged N-directed slab dip. This is also observed by other authors who interpret a vertically continuous Adriatic slab down to at least 400 km depth (Zhao et al., 2016; Hua et al., 2017). Mitterbauer et al. (2011) observe a more vertically oriented anomaly compared to the Lippitsch et al. (2003) model, hence arguing that the apparent dip direction is no reliable indication of Adriatic subduction. Dando et al. (2011) infer that the deepest part of the slab can be followed almost continuously into the deep seated high-velocity anomaly below the

Pannonian Basin that they interpret as remnant of the subducted European plate. Hence the imaged slab would rather be of European origin.

In order to assess the likelihood of these different scenarios we discuss below: (1) dip direction of the slab (2) continuity of the imaged slab with either the Adriatic or European plate and Moho (3) lateral discontinuity between central and eastern Alpine slabs, (4) vertical continuity of the slab, (5) geological constraints on the amounts of collisional shortening.

(1) Slab dip: the Alpine slabs are mostly sub-vertical, with dipping angles between 70 and 90° (Lippitsch et al., 2003; Koulakov et al., 2009; Dando et al., 2011; Mitterbauer et al., 2011; Zhao et al., 2016; Hua et al., 2017; Kästle et al., 2018). The resolution limit and the geometry of the ray paths may furthermore introduce artifacts that bias the apparent dipping angle. If the entire slab is taken into account, a slight northward dip is visible in many sections through the eastern Alps (Lippitsch et al., 2003; Mitterbauer et al., 2011; Zhao et al., 2016; Hua et al., 2017). Along the Dinarides-Alps transition, the apparent dip angle may

be influenced by imaging northern Dinaric and eastern Alpine slabs at the same time, thus enhancing the northeastern dip direction of the anomaly (Fig. 2D).

(2) Moho structure: attributing the high-seismic-velocity, slab-like anomaly under the eastern Alps to either the European or Adriatic plate is difficult as teleseismic body-wave tomography suffers from smearing along ray paths in the uppermost mantle and is strongly dependent on crustal velocity corrections. Additionally, the Moho structure itself suffers from uncertainty in the

eastern Alps and gives no clear hint on which plate is on top (e.g., Lüschen et al., 2004; Spada et al., 2013). The most recent work along the EASI dense-station profile (along 13° longitude) suggest that the Adriatic Moho goes below Europe, although the uncertainty is shown to be high (Hetényi et al., 2018b). Thus we do not find a reliable link between either European or Adriatic Moho and the slab anomalies below (Fig. 2C,D), and the surface-wave tomography does not provide any clear indications either (Fig. 1).





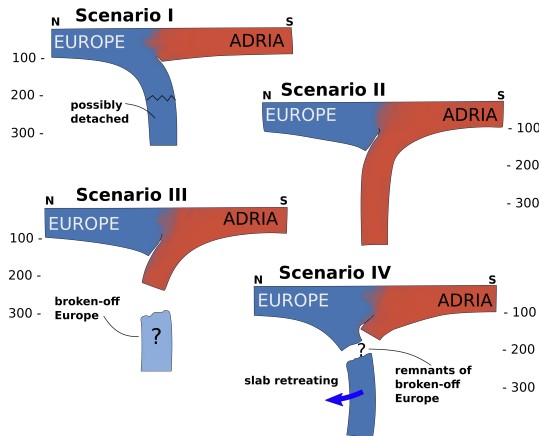

**Figure 3.** Schematic illustration of the four discussed scenarios. The sections cross the eastern Alps approximately along profile C in Figs. 1 and 2.

(3) Lateral continuity: the absence of a lateral continuity between central and eastern Alpine slabs is observed in all models (Lippitsch et al., 2003; Koulakov et al., 2009; Mitterbauer et al., 2011; Zhao et al., 2016; Hua et al., 2017; Kästle et al., 2018). There is either a horizontal slab gap (Fig. 2 Lippitsch et al., 2003) or a northward step of the anomaly (Fig. 2 Koulakov et al., 2009; Mitterbauer et al., 2011; Zhao et al., 2016). This discontinuity coincides spatially with the sinistral Giudicarie fault (e.g.,

Zhao et al., 2016).

(4) Vertical continuity: the eastern Alpine slab terminates sharply at 250 km depth in the model of Lippitsch et al. (2003, Fig. 2C,D), but all other models image also deeper anomalies (Fig. 2C,D, S10–S12; Koulakov et al., 2009; Mitterbauer et al., 2011; Dando et al., 2011), possibly in continuity to the shallow slab (Zhao et al., 2016; Hua et al., 2017). East of 14° longitude, several models indicate that the shallow part of the slab disappears and only a deep slab below 250 km is visible (Koulakov

et al., 2009; Mitterbauer et al., 2011; Zhao et al., 2016; Hua et al., 2017). Our surface-wave model does not cover the deeper structures, but shows a weak but continuous fast anomaly in the top 200 km (Fig. 1).

(5) Shortening constraints: the total convergence since 25 Ma is estimated in the order of 190 km in the eastern Alps by assuming a 20° counter-clockwise rigid-body rotation of Adria (Ustaszewski et al., 2008). However, collisional shortening estimates provide minimum estimates of 75 km in the European plate (Rosenberg et al., 2017) and 50 km in the Adriatic plate

(Schönborn, 1999; Nussbaum, 2000).

*Based on the presented arguments we discuss four scenarios for the eastern Alps, three of which are taken from the literature and a fourth one in order to reconcile the presented findings (Fig. 3):*

**Scenario (I): Continuous European slab down to at least 250 km**

This scenario represents a situation analogue to the one inferred for the central Alps and was the accepted one before the work of Lippitsch et al. (2003) started a discussion on a possible subduction of Adria under the eastern Alps. This scenario can





explain the apparent vertical continuity of the slab, reaching down to depths $\geq 350\ km$ as shown in several models (Figs. 2, S10–12, Koulakov et al., 2009; Dando et al., 2011; Mitterbauer et al., 2011; Zhao et al., 2016). Only Lippitsch et al. (2003) show a shorter slab down to 250 km, which might suggest that the slab is broken-off at this depth (Fig. 2).

However, scenario (I) cannot explain the above discussed lateral discontinuity. The surface-wave model shows a significant
difference in anomaly strength when comparing central and eastern Alps (Fig. 1), which we is not only due to the difference in station coverage from west to east (Kästle et al., 2018), suggesting that the European slab is strongly thinned or detached in the uppermost mantle layer of the eastern Alps. This becomes also evident from the body-wave models, east of 14° longitude, where there is no high-velocity anomaly in the top 250 km (Figs. 2D, S12, Koulakov et al., 2009; Dando et al., 2011; Mitter-bauer et al., 2011; Zhao et al., 2016; Hua et al., 2017). The slight northward dip of the slab in almost all models is another
argument against European subduction, although it remains ambiguous and the slab could be overturned.

### Scenario (II): Continuous Adriatic slab down to 400 km

Zhao et al. (2016), Hua et al. (2017) and (Hetényi et al., 2018b) propose that the vertically continuous and slightly northward dipping fast anomaly under the eastern Alps could represent a long Adriatic slab (Figs. 2C,D, S10–S12). We discard this possibility, because of the lack of geologic evidence of a long-lasting Adriatic subduction in the eastern Alps (Handy et al.,
2010, and references therein) and the aforementioned estimates of collisional shortening that are by far too small to explain a continuous Adriatic slab down to the mantle transition zone.

### Scenario (III): Adriatic subduction down to 250 km

This scenario (Lippitsch et al., 2003) requires a European slab break off and the onset of Adriatic subduction, or an extension of the Adriatic subduction from the Dinarides into the eastern Alps. Adriatic subduction would have been preceded by European
slab tear and eastwards retreat, linked to the opening of the Pannonian Basin and the lateral movement along the Giudicarie Fault, which opened up the necessary space for the Adriatic slab (Schmid et al., 2004; Handy et al., 2015). This scenario can thus explain the lateral slab discontinuity from central to eastern Alps, evidenced in all tomographic models, approximately coinciding spatially with the Giudicarie Fault. It remains unclear whether such a European slab tear is related to the 32-30 Ma magmatism (Rosenberg, 2004) or whether a younger break-off is necessary to enable post-20 Ma Adriatic subduction (Handy
et al., 2015).

A major problem with this scenario is the insufficient shortening on the Adriatic plate in the eastern Alps of only 50 km (Schönborn, 1999), as inferred from balancing of cross sections (Nussbaum, 2000). In order to explain a 200 km long Adriatic slab from the Lippitsch et al. (2003) model, Handy et al. (2015) propose that the slab may have deformed and stretched, driven by suction from the broken-off European slab, or that the tomographic image shows an amalgamation of European and Adriatic
slab. The insufficient shortening of the Adriatic crust is even more difficult to explain with respect to the other models that show a slab length of 300 km and more (Fig. 2, S10–S12).

### Scenario (IV): Detached European slab, shallow subduction of Europe and Adria

The discrepancy between minimum shortening estimates in the southern Alps and imaged slab lengths under the eastern Alps corresponds to $\geq$100 km for the model of Lippitsch et al. (2003) and $\geq$200 km for the other models. This can hardly be ex-
plained by Adriatic slab stretching or smearing, but could be explained by subduction of both European and Adriatic slabs into




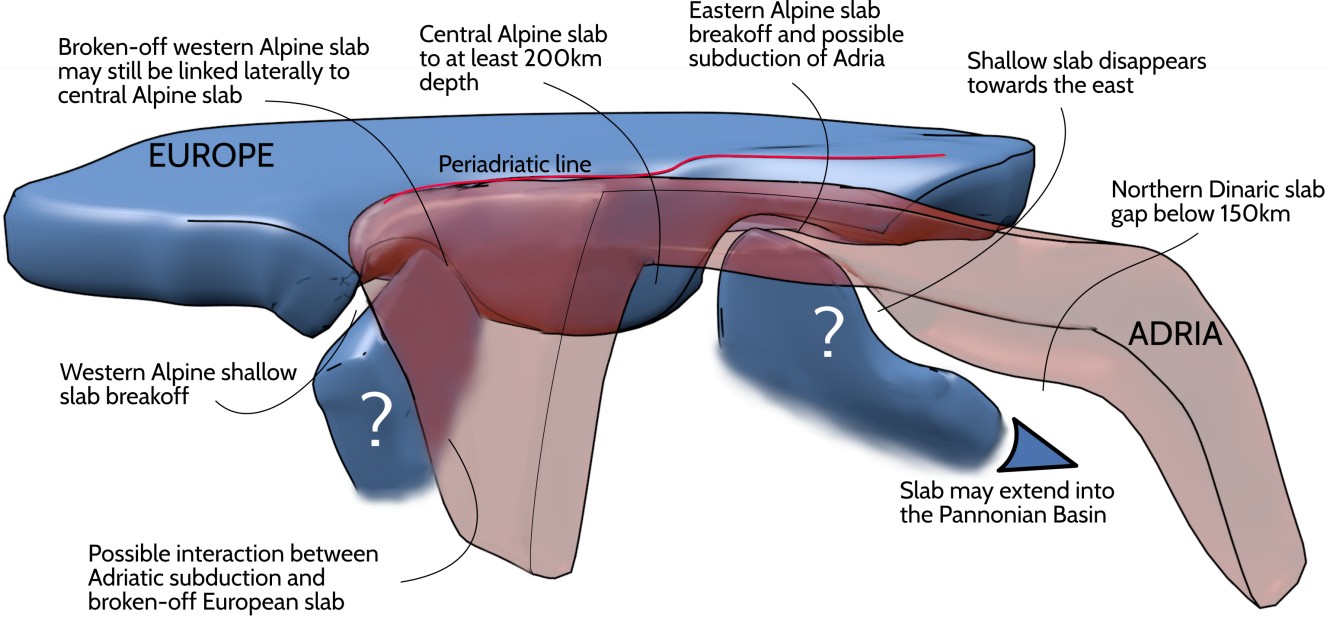

**Figure 4.** Schematic model of a possible configuration of European and Adriatic plates as presented in Scenario (IV). Question marks indicate very speculative locations of the European broken-off slabs. Most of the Apenninic slab is cut out from the sketch to improve visual clarity.

the upper mantle (Fig. 4). Considering the inferred, very shallow level of detachment of the European slab, as derived from the surface-wave model (Fig. 1) and the apparent continuity of the slab in many models, a young break-off age, significantly younger than the one inferred to cause the Periadriatic magmatism at 32–30 Ma, is required. Such a detachment would propagate in a wedge-like fashion from the east, possibly related to the eastward escape of the Carpathians (Schmid et al., 2004) and similar to the model of Qorbani et al. (2015) based on their SKS-splitting results. This model is in agreement with the vertical slab gap in the top 250 km, east of 14° longitude (Fig. 2D, S12, Koulakov et al., 2009; Mitterbauer et al., 2011; Zhao et al., 2016; Hua et al., 2017). A small amount of northward directed subduction of Adria filling the gap left by the broken-off European slab could explain why many tomographic models show a continuous vertical anomaly in the uppermost mantle layer between 12° and 14° longitude. Hence, the combination of remnants of Europe and onsetting Adriatic subduction could explain the weak, high-velocity anomaly in the surface-wave model.

A slight northward dip of the eastern Alpine slab is observed in most models, both in the shallow and in the deep parts of the slab. This may be caused by a retreat of the European slab (Kissling and Schlunegger, 2018) or northward directed asthenospheric flow under the northern Adriatic plate (e.g., Vignaroli et al., 2008), or by imaging a combination of shallow Adriatic and deep European slab (Fig. 3). A northward retreat of Europe would also explain the northward step of the anomaly from central to eastern Alps as observed in some models (Zhao et al., 2016; Hua et al., 2017). The deep, broken-off part of the European slab may be linked to the slab remnants in the mantle transition zone below the Pannonian basin (Dando et al., 2011, Fig. 4).





We assume that there is a structural transition along strike of the eastern Alps, related to the proposed wedge-like detachment: Adriatic subduction may play no or only a very minor role in the westernmost part (Tauern window) where the Adriatic slab should not be significantly longer than the inferred amounts of shortening of ∼50 km (Fig. 3). Further east, parts of the Dinaric slab may have propagated laterally, due to the counter-clockwise rotation of Adria, into the eastern Alps, thus showing a longer

Adriatic slab there and approaching Scenario (III) towards the eastern end of the eastern Alps. This would explain why the northward dip is clearest at the Alps-Dinarides transition, but less obvious close to the TRANSALP cross-section along approx. 12° (Figs. 2, 4, S10–S12).

The fast anomaly under southern Germany, at around 150 km depth in the surface-wave model (EANA, Fig. 1C) and the one of Lippitsch et al. (2003) (Fig. 2) may be related to a European slab retreat, comparable to the Vrancea slab which is located under

the Carpathian foreland (Knapp et al., 2005). The direction is compatible with the sinistral offset along the Giudicarie fault, the location at 200 km north of the eastern Alps is, however, quite far compared to the estimated 70 km of lateral movement along the fault (Scharf et al., 2013). The relation of the foreland anomaly to the eastern Alpine subduction may improve our understanding of the processes in the eastern Alps.

Similar to the western Alps, it will be necessary to improve the tomographic images in the eastern Alps and the northern

Dinarides especially in the top 150 km, thus showing the crust-mantle transition and giving the possibility of attributing the deeper anomalies to either the European or Adriatic plate. This is linked to having better spatial and temporal constraints for the slab break-off(s) and testing the hypothesis of northward-directed European slab retreat. We expect that further insight will be provided by the AlpArray experiment (Hetényi et al., 2018a).

**6 Conclusions**

Our comparison of regional high-resolution tomography models shows that some of the body-wave models differ significantly in shape and length of the imaged slabs. As a consequence there is no consistent model of the Alpine subduction and slab geometries, yet. In order to understand detachments at lithospheric level, good resolution in the entire upper mantle up to the Moho is required. Teleseismic body-wave models are often limited in the uppermost 150 km due to steep incident angles of

teleseismic rays, whereas the shear-velocity model from surface-wave recordings provides a better resolution.

In the western Alps, we favor the scenario of a European slab which is detached slightly below the lithosphere. Hence, the absence of slab pull may cause the recent uplift despite the absence of convergence. At greater depth, tomographic models cannot yet clearly distinguish between the detached European slab and Adriatic subduction under the northern Apennines that

may be merged into one slab.

Concerning the eastern Alps, we object the interpretation of a continuous Adriatic slab in the entire upper mantle, however, shallow Adriatic subduction cannot be ruled out. We suggest that the amount of subduction should be consistent with southern Alpine shortening estimates of ≥50 km and may only increase towards the northern Dinarides where the Adriatic slab is known
to be of 150 km length. A European slab break-off could have propagated from the east related to the eastward migration of the Carpathians. Adria may have partially filled this gap. We summarize these findings in a new, fourth scenario, which will stimulate the discussion on the eastern Alpine slab geometries and provides a more evolved hypothesis to be further tested with the AlpArray (Hetényi et al., 2018a) experiment, ideally with a combination of methods to combine the strength of both body-

and surface-wave tomography.

*Data availability.* All data used in the surface-wave study is stored in the EIDA archive (http://www.orfeus-eu.org/eida). The final surface-wave model is freely accessible, including Python routines for plotting (http://hestia.lgs.jussieu.fr/ boschil/downloads.html). The body-wave tomographic models are available on request via the original authors.

*Author contributions.* EK created the model comparison and prepared the MS with contributions from all co-authors. CR contributed to the interpretation and discussions on the Alpine structures. EK and LB created the surface-wave tomographic model. NB contributed to the discussion on geological and tectonic context of the Alpine subduction. TM and AE contributed with an earthquake surface-wave dataset and assisted both the technical and interpretative discussions.

*Competing interests.* The authors declare that they have no conflict of interest.

*Acknowledgements.* We acknowledge the two reviewers, C. Piromallo and M. Malusà, for their helpful comments and suggestions. We also thank A. Paul, B. Dando, Y. Hua, I. Koulakov, U. Mitterbauer, R. Lippitsch and C. Piromallo for providing the tomographic models. We are grateful to all the network operators providing data to the EIDA archive. Graphics were created with Python Matplotlib, Blender and Inkscape. EK has received funding from the French GRNE graduate school (ED-398) and the German Science Foundation (SPP-2017, Project Ha 2403/21-1). LB has received funding from the European Union's Horizon 2020 research and innovation program under the Marie

Sklodowska-Curie grant agreement No. 641943.



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
