# Peer review of "Slab Break-offs in the Alpine Subduction Zone"

_Solid Earth, 2019_

## Referee Comment (RC1) · Barbara Romanowicz (Referee) · 21 Jun 2019

The idea behind this paper is to combine results from surface wave tomography on the one hand and from body wave tomography of the upper mantle beneath the Alps to choose among possible proposed scenarios of tectonic evolution of the region after continental collision, involving the fate of several subducted slabs.
The authors argue that by combining the two types of results, they take advantage of better resolution of surface waves in the shallow layers (<200 km depth), and additional constraints of body waves in the deeper upper mantle layers.

The main issue I have with the paper in its current form is the presentation: the authors start from the idea of combining the results of surface wave and body wave tomography, trusting the surface wave tomography better at shallow depth, but they don't really allow us to easily judge what happens when you do that: the surface wave and body wave models are presented at different scales (in particular in the depth direction) and there is no effort to adjust the color schemes between the two types of models. In particular, if I understand it correctly, the averages at a given depth taken out before plotting are not the same in the surface wave and body wave models: surface wave images as presented with respect to PREM, whereas the body wave models, by construction, are presented with respect to the regional average. It would therefore make sense to remove the regional average from the surface wave models for a comparison with the other ones. This would actually help visualize small perturbations that are currently hidden because the surface wave images are biased to blue colors in this region of convergence.

What would be very helpful is to show composite cross-sections with the surface wave model at the top, truncated at some depth (150 or 200 km?) followed by the respective body wave models (see figure 1 below where I have attempted to illustrate this concept for sections B). - You could also show, separately, comparisons of the surface wave and body wave models in the shallow parts to better visualize the compatible elements of the models. With a little more annotations of specific features in those cross-sections, it would be much easier and faster for the reader to follow the text and therefore judge the proposed interpretation, which I find very hard to do as presented. And it would be consistent with the main idea behind the paper, which is to combine the deep structure from body waves with the shallower structure from surface waves.

details:

page 7 line 24: "eastward increase", do you mean "decrease" ?

page 8 ,lines 19-21: this sentence needs pointing to specific features on one of the figures, otherwise it is hard to evaluate. More generally, better guiding the reader as to which features are discussed on which figures in the Discussion section would be helpful (for example putting more annotations on the cross-sections which would be referred to in the text.

page 8 line 30: "The inferred amount of shortening...". How do you infer that quantitatively (i.e. what rates of slab sinking ?

[Figure]

Figure 1.

---

## Referee Comment (RC2) · Helle Pedersen (Referee) · 24 Jun 2019

Review of Slab Break-offs in the Alpine Subduction Zone by Emanuel D. Kästle, Claudio Rosenberg, Lapo Boschi, Nicolas Bellahsen, Thomas Meier, and Amr El-Sharkawy

The manuscript compares different body-wave tomography studies of the Alps, which have different geodynamical interpretations. They use a recent surface wave tomography from Kästle et al. (JGR 2018)as a basis for discussion of these interpretations. The authors conclude on the presence of slab break off in the western and eastern Alps, and a continuous slab in the central Alps.

I have some major comments to the manuscript.

1. The first one is that the discussion does not bring much new on the table as compared to the Kästle et al. tomography (JGR 2018). The different tomographies and their interpretation are present in various manuscripts, and as I see it, the discussion of the present MS is already quite similar to the one in Kästle et al., 2018. For an in-depth discussion, the tomographies, including the surface wave tomography, would need to be transposed to similar scales and the Kästle et al. model should be added to Figure 2. I have for my own use, made a composite figure that combines Figures 1 and 2. It would seem at a first glance that the Kästle model overall has more agreement with the Zhao et al. model and/or the Koulakov model than with the Lippitsch et al. model – but it is really difficult to compare when the regional average at each depth has not been subtracted at each depth.

2. The second issue concerns the resolution of the surface wave models. The Alpine slab geometry is fully 3D, and in the western part of the Alps it may well be the most complex mantle geometry in the world at such a small scale. Indeed the structures are laterally small, and the crust particularly complex because of the Ivrea body and the proximity of the very deep Po Plain. At ~100 km depth, the wavelengths used are of the order of 400km (periods of approx. 100s). Care should therefore be taken at interpretation of spatially narrow (~50km-100km) areas of velocity reductions at this depth. I read the original Kästle (JGR 2018) paper and there are indeed checkerboard tests, albeit at much shorter periods. If we assume great-circle propagation, I would assume that checkerboard tests would perform well also at long periods, however these tests don't take into account the very complex wave propagation across the Alps. There is quite a lot of research that has taken place to better understand the complexity of surface wave propagation in heterogeneous structures, and even though the large amount of data used helps to improve resolution, it is optimistic to interpret anomalies as small as 10%-25% of the wavelength. Using dedicated small scale arrays is one option, but a limit of 10%-20% of the wavelength still applies (see for example Bodin et al., 2008). Also, recent work by Kolinsky et al. demonstrate that the surface wave propagation in the greater alpine area is complex, also at long periods. While these problems can probably be neglected for relatively large scale structures, they cannot be ignored for small scale structures, which should consequently be treated with utmost care and possibly not be interpreted.

3. The Ivrea body makes all types of tomography very difficult indeed, as errors on the very complex 3-D crustal model can leak into the mantle in various forms. A good example of such problems is Beller et al. (GJI, 2018) who observe an anomaly which could be the continuation of the European crust, or alternatively low velocity mantle material, indicating slab breakoff. With crustal material that extends to at least 80km, surface wave inversions may additionally have crust/mantle tradeoffs that go as deep as 100-120km.

4. In the eastern Alps, the size of the structures is bigger, so better resolved. The slab breakoff is in that case attributed to a decrease of positive velocity anomaly at approximately 100km

depth, decrease that is found to a larger or smaller extent in many parts of the model. What I don't quite understand is why this velocity decrease is taken as a slab breakoff, when such a decrease is present, but not interpreted, in other parts of the model. Perhaps this can be explained by further text – it may be an issue of 3D geometry only? At a minimum the reader needs more help.

5. The Kästle model (JGR 2018) could at that time not take advantage of the more recent models which include AlpArray data, but a re-inversion using an updated and more detailed crustal model would at this point be a valuable addition to the 2018 article. As an example, the interpretation of surface wave dispersion by Lyu et al. (GJI, 2017), using a very detailed crustal model and higher resolution due to the use of a set of dense networks, does not indicate a slab breakoff. Note that the Lyu et al. paper made the transposition of surface wave data onto a body wave tomography type representation and would make for an interesting comparison for this MS.

Minor comments

a. I disagree that the uplift is due to slab breakoff. Indeed, a recent article by Sternai et al. (Earth Science reviews 2019) demonstrates convincingly that the uplift has too small lateral extent to be explained by any of the present models. Note that this MS is recent so may not have been known to the authors.

b. If possible, it should be made easier to compare figure 1s and 2, in terms of the denominations. For people not very aware of Alpine 3D geometries, it is quite hard to figure how to compare items with different names. It will in any case help to combine the models of Figures 1 and 2, but aligning vocabulary might be useful?

**Attempt at composite image Figures 1 and 2**

---

## Author Comment (AC1) · 19 Jul 2019

Dear Barbara Romanowicz,

Dear Editors,

we appreciate the helpful comments on our manuscript which we have carefully read and taken all into account in the revised version. We re-structured Figures 1 and 2 and created a new Figure (3) to ensure better comparability between the shown models. We show now the surface-wave model in a side-by-side comparison with the body-wave models and also show it on top of the body-wave models, including additional annotations and labels to guide the reader. We have included a point-by-point response that you find below with the original comments in bold letters. We also attached the

manuscript with the highlighted changes.

**Review of Kastle et al.**

**The idea behind this paper is to combine results from surface wave tomography on the one hand and from body wave tomography of the upper mantle beneath the Alps to choose among possible proposed scenarios of tectonic evolution of the region after continental collision, involving the fate of several subducted slabs.**

**The authors argue that by combining the two types of results, they take advantage of better resolution of surface waves in the shallow layers (<200 km depth), and additional constraints of body waves in the deeper upper mantle layers.**

**The main issue I have with the paper in its current form is the presentation: the authors start from the idea of combining the results of surface wave and body wave tomography, trusting the surface wave tomography better at shallow depth, but they don't really allow us to easily judge what happens when you do that: the surface wave and body wave models are presented at different scales (in particular in the depth direction) and there is no effort to adjust the color schemes between the two types of models. In particular, if I understand it correctly, the averages at a given depth taken out before plotting are not the same in the surface wave and body wave models: surface wave images as presented with respect to PREM, whereas the body wave models, by construction, are presented with respect to the regional average. It would therefore make sense to remove the regional average from the surface wave models for a comparison with the other ones. This would actually help visualize small perturbations that are currently hidden because the surface wave images are biased to blue colors in this region of convergence.**

All tomographic images in the present manuscript (and supplementary material) are presented without any vertical or horizontal exaggeration, so that there is no distortion

of structures when comparing the surface- to the body-wave models. We consistently use the same color model but we use different scaling in the color bars. This is necessary, because the absolute velocity of the anomalies varies considerably between models and we want to make the models as comparable as possible. In the revised version we subtracted the 1D average from the surface-wave model to show relative velocity deviations from this regional average instead of PREM. This guarantees that all models are shown with respect to their individual regional average. In order to facilitate comparisons, we re-structured the manuscript so that we show the surface-wave model directly next to the body-wave models in the new Figures 1 and 2.

**What would be very helpful is to show composite cross-sections with the surface wave model at the top, truncated at some depth (150 or 200 km?) followed by the respective body wave models (see figure 1 below where I have attempted to illustrate this concept for sections B). - You could also show, separately, comparisons of the surface wave and body wave models in the shallow parts to better visualize the compatible elements of the models. With a little more annotations of specific features in those cross-sections, it would be much easier and faster for the reader to follow the text and therefore judge the proposed interpretation, which I find very hard to do as presented. And it would be consistent with the main idea behind the paper, which is to combine the deep structure from body waves with the shallower structure from surface waves.**

We re-structured Figures 1  2 to show the different models directly next to each other. We added more annotations to the cross-sections, such as denominations of the mountain ranges at the surface and use consistently the same labels in all cross sections to make it easier for the reader to follow. In the new Figure 3 we plot, as suggested, the surface-wave model on top of the body-wave models. Additionally, we highlight some structural continuities that are discussed in the manuscript. We also added more Figure references and specific references to the labels in the figures in several parts of the text.

**details:**

**page 7 line 24: "eastward increase", do you mean "decrease" ?**

Yes, decrease would have been correct. Because the sentence was confusing in the original MS we changed it to "Underneath the eastern Alps all models show a vertical to slightly northward dipping slab (Fig. 2C,D, S10–S12, Lippitsch et al., 2003; Piromallo and Morelli, 2003; Koulakov et al., 2009; Dando et al., 2011; Mitterbauer et al., 2011; Zhao et al., 2016; Hua et al., 2017). This northward dip is most clearly expressed in the model of Lippitsch et al. (2003), who image the slab down to a depth of about 250 km (Fig. 2D)."

**page 8 ,lines 19-21: this sentence needs pointing to specific features on one of the figures, otherwise it is hard to evaluate. More generally, better guiding the reader as to which features are discussed on which figures in the Discussion section would be helpful (for example putting more annotations on the cross-sections which would be referred to in the text.**

We added more references in the text, also pointing to specific figure labels in the revised version, such as "(Figs. 2A, 3A, label EU/AD)".

**page 8 line 30: "The inferred amount of shortening...". How do you infer that quantitatively (i.e. what rates of slab sinking ?**

This sentence was unclear, the inferred amount of shortening is in this case only based on the shortening estimates that are available from surface-geological studies. We changed the sentence to: "The collisional shortening estimates for the western Alps range between 30 – 150 km, when taking the shortening of the European units and 50 km of potential underthrusting of Europe into account (Bellahsen et al., 2014; Schmid et al., 2017; Rosenberg et al., 2019). The inferred amount of shortening from these works is lowest in the south, increases to about 100 km along our cross-section A (Fig. 1) and reaches its maximum in the north at the transition to the Central Alps."

Please also note the supplement to this comment:
https://www.solid-earth-discuss.net/se-2019-102/se-2019-102-AC1-supplement.pdf

---

## Author Comment (AC2) · 19 Jul 2019

Dear Helle Pedersen,

Dear Editors,

we appreciate the helpful comments on our manuscript which we have carefully read and taken all into account in the revised version. We re-structured Figures 1 and 2 and created a new Figure (3) to ensure better comparability between the shown models. We show now the surface-wave model in a side-by-side comparison with the body-wave models and also show it on top of the body-wave models, including additional annotations and labels to guide the reader. We have included a point-by-point response that you find below with the original comments in bold letters. We also attached the

manuscript with the highlighted changes.

**Review of Slab Break-offs in the Alpine Subduction Zone by Emanuel D. Kästle, Claudio Rosenberg, Lapo Boschi, Nicolas Bellahsen, Thomas Meier, and Amr El-Sharkawy**

**The manuscript compares different body-wave tomography studies of the Alps, which have different geodynamical interpretations. They use a recent surface wave tomography from Kästle et al. (JGR 2018)as a basis for discussion of these interpretations. The authors conclude on the presence of slab break off in the western and eastern Alps, and a continuous slab in the central Alps.**

**I have some major comments to the manuscript.**

**1. The first one is that the discussion does not bring much new on the table as compared to the Kästle et al. tomography (JGR 2018). The different tomographies and their interpretation are present in various manuscripts, and as I see it, the discussion of the present MS is already quite similar to the one in Kästle et al., 2018. For an in-depth discussion, the tomographies, including the surface wave tomography, would need to be transposed to similar scales and the Kästle et al. model should be added to Figure 2. I have for my own use, made a composite figure that combines Figures 1 and 2. It would seem at a first glance that the Kästle model overall has more agreement with the Zhao et al. model and/or the Koulakov model than with the Lippitsch et al. model – but it is really difficult to compare when the regional average at each depth has not been subtracted at each depth.**

The work of Kästle et al. (2018) was aimed at presenting the surface-wave model of crust and mantle and discuss its implications for existing geodynamic interpretations. However, the present MS has a much broader scope and gives a review of published models alongside explanations of main methodological differences to provide a state-of-the art discussion that can also be useful for non-tomographers. We are convinced

that the a constructive continuation of the discussion on the Alpine slab structures, which has been ongoing for around 15 years now, can only be done if an overview of concurring scenarios and inconsistencies of different published interpretations (including the ones of Kästle et al., 2018) is presented. The reviewer is right that the best comparability between models is guaranteed when the sections are shown next to each other in the same figure (as we already did in the supplementary material). We therefore re-structured the Figures 1 and 2 and added a new Figure 3 where the surface-wave model is always shown next-to or above the body-wave models. As suggested by both reviewers, we subtract the 1D average model from the surface-wave model in all presented figures. Therefore, in the revised version, all models are plotted with respect to their individual average.

**2. The second issue concerns the resolution of the surface wave models. The Alpine slab geometry is fully 3D, and in the western part of the Alps it may well be the most complex mantle geometry in the world at such a small scale. Indeed the structures are laterally small, and the crust particularly complex because of the Ivrea body and the proximity of the very deep Po Plain. At 100 km depth, the wavelengths used are of the order of 400km (periods of approx. 100s). Care should therefore be taken at interpretation of spatially narrow ( 50km-100km) areas of velocity reductions at this depth. I read the original Kästle (JGR 2018) paper and there are indeed checkerboard tests, albeit at much shorter periods. If we assume great-circle propagation, I would assume that checkerboard tests would perform well also at long periods, however these tests don't take into account the very complex wave propagation across the Alps. There is quite a lot of research that has taken place to better understand the complexity of surface wave propagation in heterogeneous structures, and even though the large amount of data used helps to improve resolution, it is optimistic to interpret anomalies as small as 10%-25% of the wavelength. Using dedicated small scale arrays is one option, but a limit of 10%-20% of the wavelength still applies (see for example Bodin et al., 2008). Also, recent work by Kolinsky et al. demon-**

**strate that the surface wave propagation in the greater alpine area is complex, also at long periods. While these problems can probably be neglected for relatively large scale structures, they cannot be ignored for small scale structures, which should consequently be treated with utmost care and possibly not be interpreted.**

We agree with the reviewer that the resolution is an issue in the surface-wave model with increasing depth, and we explain this in more detail in the revised manuscript: "The positions of the anomalies are subject to increasing uncertainty with depth, meaning that the surface-wave model is not well suited to estimate the slab dip. This may cause some differences in the anomaly positions with respect to the body-wave models. Tests have shown that the size of the anomaly should not be smaller than ∼25% of the wavelength to be resolvable (Bodin et al., 2008). Surface waves that are most sensitive to structures at 100 km depth have a period of around 60 s (Smith et al., 2004) and an average wavelength of 270 km (at 200 km the period becomes 150 s and the wavelength 675 km). This means that smearing and weakening of the imaged anomalies is observed at depth greater than 100 km, and it is not only caused by the large wavelengths but also by the reduction in data coverage at long periods in the dataset (Kastle et al., 2018). Complexities in the propagation path of surface waves can also affect the uncertainty in the deeper part of the model where it is largely constrained by earthquake data (Kolinsky et al., 2019)." The smallest interpreted structure in the manuscript is probably the western Alpine slab break off. We have revised this paragraph, writing that "From the contrast to the central Alpine structure and the missing vertical continuation to depth, we conclude that the European slab underneath the western Alps is not as thick and continuous as under the central Alps and may have broken off. The reduction of shear velocities is observed underneath the entire western Alpine area as shown in Figure 1 and is therefore still within a 50% wavelength threshold. The geometrical details of the proposed break off can, however, not be resolved with surface waves alone."

**The Ivrea body makes all types of tomography very difficult indeed, as errors on the very complex 3-D crustal model can leak into the mantle in various forms. A good example of such problems is Beller et al. (GJI, 2018) who observe an anomaly which could be the continuation of the European crust, or alternatively low velocity mantle material, indicating slab breakoff. With crustal material that extends to at least 80km, surface wave inversions may additionally have crust/mantle tradeoffs that go as deep as 100-120km.**

In the revised version of the manuscript we added the sentence "The observed reduction in anomaly strength under the western Alps may, however, be influenced by subduction of crustal material as shown by Zhao et al. (2015)". We explain also the uncertainties from the surface-wave interpretations more clearly: "From the contrast to the central Alpine structure and the missing vertical continuation to depth, we conclude that the European slab underneath the western Alps is not as thick and continuous as under the central Alps and may have broken off. The reduction of shear velocities is observed underneath the entire western Alpine area in Figure 1 and is therefore still within a 50% wavelength threshold. The geometrical details of the proposed break off can, however, not be resolved with surface waves alone." We show that the break-off interpretation is still subject of significant uncertainty (p.9): "A slab break-off would agree with the models of Beller et al. (2017, full-waveform modeling) and Lippitsch et al. (2003, body-wave tomography), who locate the gap between 80 and 150 km depth. However, several other tomographic models image a continuous slab down to at least 250 km depth (Koulakov et al., 2009; Zhao et al., 2016; Hua et al., 2017; Lyu et al., 2017), implying that there was no break-off at all in the western Alps."

**4. In the eastern Alps, the size of the structures is bigger, so better resolved. The slab breakoff is in that case attributed to a decrease of positive velocity anomaly at approximately 100km depth, decrease that is found to a larger or smaller extent in many parts of the model. What I don't quite understand is why this velocity decrease is taken as a slab breakoff, when such a decrease**

**is present, but not interpreted, in other parts of the model. Perhaps this can be explained by further text – it may be an issue of 3D geometry only? At a minimum the reader needs more help.**

We base this interpretation on the difference between central and eastern Alps, which we explain more clearly now: "Given that a subduction of the European plate affected both central and eastern Alps until at least 35 Ma ago (e.g., Handy et al., 2015, and references therein), the reduction of anomaly strength between the two Alpine domains suggests that they must have experienced a different evolution in more recent times (Fig. 1 CAA to EAA): the European slab under the eastern Alps is either thinned, or even broken off. The high-velocity anomaly under the eastern Alps at 60 – 200 km depth may also correspond to an Adriatic slab, which generally shows a lower velocity anomaly strength underneath the Apennines in our model." All following discussions on the eastern Alpine slab and a potential break off also include other tomographic results in which we see a lateral discontinuity between central and eastern Alps and a reduced anomaly strength in the top 150 km.

**5. The Kästle model (JGR 2018) could at that time not take advantage of the more recent models which include AlpArray data, but a re-inversion using an updated and more detailed crustal model would at this point be a valuable addition to the 2018 article. As an example, the interpretation of surface wave dispersion by Lyu et al. (GJI, 2017), using a very detailed crustal model and higher resolution due to the use of a set of dense networks, does not indicate a slab breakoff. Note that the Lyu et al. paper made the transposition of surface wave data onto a body wave tomography type representation and would make for an interesting comparison for this MS.**

We estimate that the effect of the additional ambient-noise data would at this point only provide a minor contribution to the mantle structures. Preliminary results of ongoing work on the AlpArray data (e.g. Kästle et al., 2019, EGU Abstracts, Vol. 21 EGU2019-8661) indicate that the main differences are visible in border regions where formerly

no stations where placed and also at shallow depth, i.e. at short periods. In contrast, the lower crustal structures do not show significant differences in the phase-velocity maps, which implies that the effect on the mantle structures is expected to be rather small from ambient-noise data alone. Moreover, the mantle structures in the shown surface-wave model are largely constrained by the earthquake two-station data. To my knowledge, no one is currently working on an updated data set which would require a lot of work. The influence is also not that obvious given the long periods of the data. Our MS is aimed at reviewing several published tomographic models and providing an in-depth discussion of different scenarios rather than creating a new model.

**Minor comments a.b.I disagree that the uplift is due to slab breakoff. Indeed, a recent article by Sternai et al. (Earth Science reviews 2019) demonstrates convincingly that the uplift has too small lateral extent to be explained by any of the present models. Note that this MS is recent so may not have been known to the authors.**

Sternai et al. (2019) discuss the different contributions to the uplift very carefully and are certainly cautious in attributing it to a slab break-off. Nevertheless, they estimate that a mantle source is likely to explain the uplift, including slab break off. For this reason we cited them already in the original version of the manuscript. In their conclusions they write: "We suggest that rock uplift rates due to the melting of the LGM Alpine ice-cap and erosion contribute up to ∼50% to the observed vertical displacement rates in the Western and Central Alps. This implies substantial contributions by convective processes (e.g., detachment of the western European slab) to the measured surface displacement rates..."

**If possible, it should be made easier to compare figure 1s and 2, in terms of the denominations. For people not very aware of Alpine 3D geometries, it is quite hard to figure how to compare items with different names. It will in any case help to combine the models of Figures 1 and 2, but aligning vocabulary might be useful?**

In order to enhance comparability, following this comment and the one of the second reviewer, we re-structured Figures 1 and 2 and added a new Figure to the manuscript. In all cross-sections we consistently use the same labels now. We also added the names of the mountain ranges so that the figures become easier to read.

Please also note the supplement to this comment:
https://www.solid-earth-discuss.net/se-2019-102/se-2019-102-AC2-supplement.pdf